# Whole body MRI in multiple myeloma: Optimising image acquisition and read times

**Saurabh Singh**[1], **Elly Pilavachi**[2], **Alexandra Dudek**[2], **Timothy J. P. Bray**[1],
**Arash Latifoltojar**[1], **Kannan Rajesparan**[2], **Shonit Punwani**[1], **Margaret A. Hall-Craggs**[1]*

**1** Centre for Medical Imaging, Division of Medicine, University College London, London, United Kingdom,
**2** Department of Radiology, University College Hospital, London, United Kingdom

\* m.hall-craggs@ucl.ac.uk

**Data Availability Statement:** All relevant data are within the paper and its Supporting Information files.

## Abstract

### Objective

To identify the whole-body MRI (WB-MRI) image type(s) with the highest value for assessment of multiple myeloma, in order to optimise acquisition protocols and read times.

### Methods

Thirty patients with clinically-suspected MM underwent WB-MRI at 3 Tesla. Unenhanced Dixon images [fat-only (FO) and water-only (WO)], post contrast Dixon [fat-only plus contrast (FOC) and water-only plus contrast (WOC)] and diffusion weighted images (DWI) of the pelvis from all 30 patients were randomised and read by three experienced readers. For each image type, each reader identified and labelled all visible myeloma lesions. Each identified lesion was compared with a composite reference standard achieved by review of a complete imaging dataset by a further experienced consultant radiologist to determine truly positive lesions. Lesion count, true positives, sensitivity, and positive predictive value were determined. Time to read each scan set was recorded. Confidence for a diagnosis of myeloma was scored using a Likert scale. Conspicuity of focal lesions was assessed in terms of percent contrast and contrast to noise ratio (CNR).

### Results

Lesion count, true positives, sensitivity and confidence scores were significantly higher when compared to other image types for DWI ($P<0.0001$ to 0.003), followed by WOC (significant for sensitivity ($P<0.0001$ to 0.004), true positives ($P = 0.003$ to 0.049) and positive predictive value ($P< 0.0001$ to 0.006)). There was no statistically significant difference in these metrics between FO and FOC. Percent contrast was highest for WOC ($P = 0.001$ to 0.005) and contrast to noise ratio (CNR) was highest for DWI ($P = 0.03$ to 0.05). Reading times were fastest for DWI across all observers ($P< 0.0001$ to 0.014).

**Funding:** This work was undertaken at UCLH/UCL, which receives funding from the Department of Health's the National Institute for Health Research (NIHR) Biomedical Research Centre (BRC) funding scheme. The views expressed in this publication are those of the authors and not necessarily those of the UK Department of Health. MHC, SP, TJPB and AL are supported by the NIHR University College London Hospitals BRC. This work was supported by CRUK/EPSRC KCL/UCL Comprehensive Cancer Imaging Centre. The funders had no role in study design, data collection and analysis, decision to publish, or preparation of the manuscript.

**Competing interests:** The authors have declared that no competing interests exist.

## Discussion

Observers detected more myeloma lesions on DWI images and WOC images when compared to other image types. We suggest that these image types should be read preferentially by radiologists to improve diagnostic accuracy and reporting efficiency.

## Introduction

Whole body MRI (WB-MRI) is a valuable tool for assessing disease in patients with multiple myeloma (MM) [1–5], and has been recommended by both the International Myeloma Working Group (IMWG) and National Institute for Health and Care Excellence (NICE) as first line imaging for the initial assessment of disease in patients with suspected MM [6,7]. Information from MRI is also incorporated into staging systems such as the Durie-Salmon PLUS staging system [8], and the number of lesions detected correlate closely with mortality [9].

To facilitate wider use of WB-MRI, efforts need to be made to improve the 'value' of WB-MRI by reducing acquisition time, reading time and overall cost [10]. A strategy to improve value of WB-MRI would be to choose sequences which provide sufficient diagnostic information in a reasonable time and remove sequences which do not add any additional diagnostic information. Recent studies have reported the utility of faster sequences in whole body protocols for MM such as Dixon MRI and diffusion weighted imaging [5,11–13]. These sequences have shorter acquisition times than conventional spin echo sequences and can provide additional functional information such as disease activity or bone marrow composition. In particular, Dixon imaging can generate four image types: in-phase (IP), out of phase (OP), water-only (WO) and fat-only (FO). These image types provide anatomical information comparable to conventional spin echo sequences but also allow the assessment of water and fat content of bone marrow and myeloma lesions. Dixon imaging is also recommended as a core component of imaging protocols and considered important in reducing false positive lesions on DWI [13]. Gadolinium-enhanced sequences have also been shown to be useful in the assessment of MM [14], and can be included into acquisition protocols [13]. Therefore, there are a number of image types in a typical WB-MRI protocol that provide diagnostic information, but it is unclear whether all sequences need to be read in order to detect bone myeloma lesions. There is a need to elucidate which sequences provide the most diagnostic information and, on that basis, eliminate less useful sequences from the acquisition and read protocols.

There are only a few studies which have examined the diagnostic performance of sequences and compared them systematically. A study by Weininger et al 2009 [14], compared conventional pre- and post-contrast spin echo sequences and a T2-weighted inversion recovery sequence (T2w-TIRM) in detecting myeloma lesions. They found that contrast enhanced T1W and the T2w-TIRM had the highest level of sensitivity [14]. A previous study by our group investigated the diagnostic performance of different non-contrast Dixon image types in detecting myeloma lesions. This study showed that FO and WO only pre-contrast images detected the most myeloma lesions compared to in phase and out of phase images [15]. No studies have directly compared the performance of unenhanced Dixon images, contrast enhanced Dixon images and DWI.

In this study, we aimed to evaluate radiologists' myeloma lesion detection rates on unenhanced fat only (FO), contrast-enhanced fat only (FOC), unenhanced water only (WO), contrast-enhanced water only (WOC) and diffusion weighted images (DWI). We hypothesised that readers would detect more true positive lesions on contrast enhanced images compared to

unenhanced images. We also hypothesised that diffusion weighted imaging would be most sensitive for detecting focal lesions.

## Materials and methods

### Subjects

Institutional review board approved this prospective study (Research Ethics Committee reference, NRES London-Bromley, 12/LO/0428) and all patients gave written informed consent.

This study prospectively enrolled thirty patients (13 males and 17 females, median age 55, age range 36–82) who were being investigated for suspected symptomatic multiple myeloma between June 2012 and September 2014. This cohort of patients has also been studied in previous publications on treatment response and detection of myeloma lesions on pre-contrast imaging [15,16]. Patients were excluded from the study if they had a history of previous malignancy or previous chemotherapy/radiotherapy, estimated GFR < 50 mL/min/1.73 m$^2$, were unable to given informed consent or had a contraindication to MRI scanning. Further investigation showed that 26 out of 30 had MM, two a had solitary plasmacytoma and one had monoclonal gammopathy of uncertain significance. The clinical and biochemical parameters for each patient are recorded in Table 1 including baseline interphase fluorescence in situ hybridisation (FISH) and genetic risk was determined according to International Myeloma Working Group recommendations [17].

### Scan acquisition

WB-MRI was carried out on a 3.0T wide-bore system (Ingenia; Phillips Healthcare, Best, Netherlands) using two anterior surface coils, a head coil and an integrated posterior coil. The

**Table 1. Patient data.**

| Patient characteristics | Number or median (range) |
|---|---|
| Age (years) | 56 (36–80) |
| Chain isotype | |
| IgG | 17 |
| IgA | 5 |
| Light chain | 4 |
| MGUS | 1 |
| Solitary plasmacytoma | 2 |
| Smouldering MM | 1 |
| ISS Stage | |
| I | 13 |
| II | 13 |
| III | 4 |
| Biochemical parameters | |
| Bone marrow percentage plasma cells | 65 (0–90) |
| Beta-2 microglobulin (mg/l) | 3.3 (1.3–11.3) |
| Albumin (g/l) | 40 (30–53) |
| Creatinine | 56 (77.5–105) |
| Genetic risk group | |
| Low/Standard Risk | 17 |
| High Risk | 9 |

Patient demographics, disease parameters and treatment. ISS, international staging system (15)

imaging protocol was comprised of coronal pre- and post-contrast modified Dixon (Dixon) acquisitions from which fat, water, in-phase and out-of-phase images were reconstructed on the scanner using a two-point method [18] (TR 3.0ms, TE 1.02–18, flip angle 15˚, slice thickness 5mm, pixel bandwidth 1992Hz, acquisition matrix 196 x 238, SENSE factor 2, number of slice 120) in addition to diffusion and post-contrast imaging covering vertex to toe using ten contiguous anatomical stations (Table 2). The Dixon images were in the coronal plane and DWI in the transverse plane. The highest b value (1000 s/mm$^2$) images were selected based on previous studies which show that the higher b value, the higher the contrast between normal and infiltrated bone marrow [14].

## Image assessment

The five individual image types including diffusion weighted images (b = 1000), unenhanced FO and WO Dixon images, and enhanced FO and WO Dixon images for all thirty patients (150 image series) were randomised and read by three readers with experience in whole body MR imaging (between three and fifteen years). In other words, the five sets of images of each individual patient were not read sequentially but in a random order. On each image series, each reader was asked to label the number of myeloma lesions present in the bony pelvis (pubis, ischium, ilium and sacrum) up to a maximum of 20. In cases where the reader assessed the disease to be diffuse or over 20 focal lesions, the patient was assigned a lesion count of 20. In addition, the readers were asked to provide a confidence score based on their degree of certainty that there were myeloma lesions in the pelvis on a 4-point Likert scale (1-no lesions, 2-indeterminate lesions, 3-likely myeloma lesions, 4-very likely myeloma lesions). The labelled images from the readers was then compared to a composite reference standard consisting of diffusion-weighted, pre- and post-contrast Dixon imaging, which had been evaluated simultaneously by a consultant radiologist with over 20 years of experience in myeloma and MR imaging. Lesions were labelled on the reference imaging as positive for myeloma if they demonstrated a combination of features that together, in the view of the experienced radiologist, made the lesion highly likely to represent myeloma. These features included abnormal marrow signal compared to background marrow (i.e. hypointense on IP and FO images, and hyperintense on WO images) and contrast enhancement and increased signal on high b value

**Table 2. Sequence parameters (15).**

| Sequence Parameters | | |
|---|---|---|
| Parameters | Dixon (pre and post contrast) | DWI (b0, 100, 300, 1000 s/mm$^2$) |
| Imaging Plane | Coronal | Transverse |
| Sequence type | Gradient echo | Single-shot spin echo with echo planar readout |
| Echo time (ms) | 1.02/1.8 | 71 |
| Repetition time (ms) | 3 | 6371 |
| Field of View (mm x mm) | 502 x 300 | 500 x 306 |
| Voxel size (mm x mm) | 2.1 x 2.1 | 4 x 4.2 |
| Number of Slices | 120 | 40 |
| Slice Thickness (mm) | 5 | 5 |
| Acquisition Matrix | 144 x 238 | 124 x 72 |
| ETL | 2 | 39 |
| Acceleration factor (SENSE) | 2 | 2.5 |
| Pixel Bandwidth (Hz) | 1992 | 3369 |
| Scan time (s) | 17 | 152 |

diffusion images. If this signal abnormality was localised on a background of normal bone marrow signal, the disease pattern was ascribed to be focal and where this was diffuse with only localised areas of normal bone marrow signal, the disease pattern was assessed to be diffuse. We did not assign a maximum lesion count for the reference standard and all individual lesions were labelled, enabling direct comparison for any lesions labelled by the readers. We also recorded whether patients had focal or diffuse disease (the diffuse classification included patients with focal-on-diffuse infiltration), based on the reference labelled imaging.

Each labelled lesion was compared with the reference standard to determine the number of true positive lesions (TP), false positive lesions (FP) (i.e. those that were incorrectly identified as lesions); and false negative lesions (FN) (these were the 'reference-standard lesions' which were not identified). This data was used to determine the mean per-set lesion count, sensitivity (TP/TP+FN), positive predictive value (TP/TP + FP) and mean confidence score.

Although other sequences were acquired during the standard whole body protocol, these were not included for assessment in the current study as these are acquired mainly for assessment of extramedullary disease (axial T2) and for measurement of response to therapy (T2, ADC and fat fraction maps before and after treatment).

## Design and statistics

The study design is illustrated in Fig 1. The statistical analysis plan was based on that used in a previous study [15]. A multilevel mixed-effects linear regression model was used to compare each lesion detection metric (lesion count, sensitivity, positive predictive value and mean confidence score), across the five image types using Stata [Stata IC Version 14.1, College Station, USA]. This analysis is designed to compare metrics between groups whilst accounting for the multiple levels within the data. Image type (i.e. FO, FOC, WO, WOC, DWI) was used as the predictor variable, and the value of the specific lesion detection metric being analysed (i.e. lesion count, sensitivity, positive predictive value or mean confidence score) was used as the outcome variable. The two levels data were clustered were at the level of 'subject' (patient) and 'observer' (reader). The same analysis was used for the two subgroups of patients who had diffuse and focal disease (as determined by the reference standard assessment).

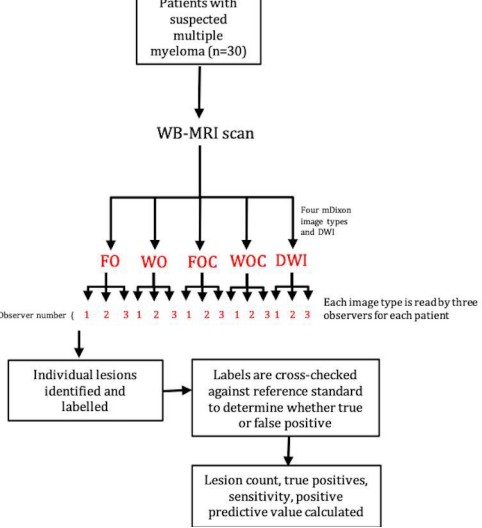

**Fig 1. Study design (15).**

**Percent contrast and contrast-to-noise ratio.** In patients with at least three focal lesions greater than 5 mm in diameter, visible on all image types, percent contrast and contrast-to-noise ratio (CNR) were calculated using a previously described method [11]. Specifically, circular regions of interest (ROIs) were drawn on focal lesions greater than 3mm in diameter and further three ROIs were placed in areas of bone marrow without focal lesions in the sacrum and iliac bones.

Percent contrast was calculated as:

$$PercentContrast = \frac{(S_a - S_b)}{(S_a + S_b)} \qquad [1]$$

where $S_a$ is the mean signal intensity of myeloma lesions and $S_b$ is the background marrow signal intensity.

Similarly, CNR was calculated as:

$$CNR = \frac{|S_a - S_b|}{\sqrt{(S_{asd} + S_{bsd})/2}} \qquad [2]$$

where $S_{asd}$ and $S_{bsd}$ are the mean within-ROI standard deviation values for myeloma lesions and background marrow respectively. A one-way analysis of variance (ANOVA) with a post-hoc Tukey Kramer multiple comparison test was used to compare percent contrast and CNR between image series.

**Patient by patient analysis.** A patient by patient analysis was undertaken after the initial analysis to determine which image types add diagnostic value. The aim of this analysis is to establish whether the observer identified any additional lesions on other image types when compared to the image type which scores the highest on the metrics examined in this study such as sensitivity, positive predictive value, true positive lesion rate and confidence.

**Reading time.** The time taken to read each image series was recorded by one of the authors. All readers were aware they were being timed and there were no specific instructions to influence reading times. All 150 image series (five image sequences for each of the 30 patients) were randomised and therefore the sequence of patients and image types were random.

## Results

Three readers read a total of 150 image series and identified 1243, 1440 and 1207 lesions respectively, compared to 1952 reference lesions. An example of a patient with a focal lesion on the five image types is shown in Fig 2. Fig 3 charts mean lesion count, sensitivity, positive predictive value and confidence for each image type and all observers. A summary of this data and true positive count for each image type for all patients (focal and diffuse disease) is given in Table 3. Sub-group analysis for focal disease only (24 patients) and diffuse disease only (6 patients) follow in Tables 4 and 5. A summary of the mean time taken for readers to read each image type for each patient is recorded in Table 6.

### Sensitivity and positive predictive value

The mean sensitivity for each image type was 0.46 for FO, 0.47 for WO, 0.46 for FOC, 0.61 for WOC and 0.72 for DWI. Sensitivity was significantly higher on the DWI images (p values from <0.001 to 0.018). WOC was the next most sensitive at detecting lesions (p values <0.001 to 0.004). There was no significant difference between WO and FO images (p = 0.909), or between FOC and FO images (p = 0.968).

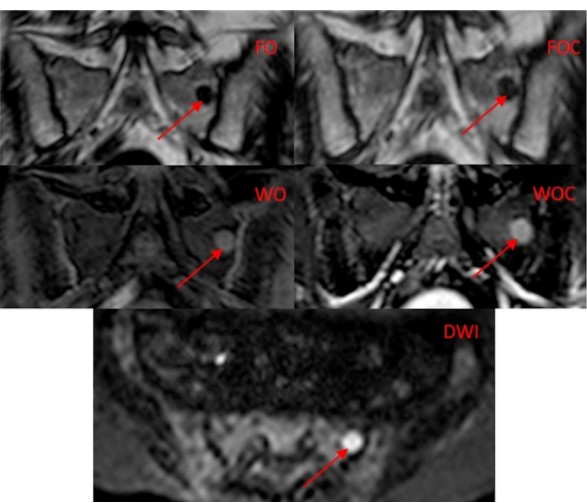

**Fig 2. Example of a focal MM lesion.** There is a focal lesion in the left hemi sacrum (arrow) on the five image types. The Dixon image types (FO, WO, FOC, WOC) are in the coronal plane and the DWI image is in the transverse plane).

The mean positive predictive values were 0.75 for FO, 0.83 for WO, 0.77 for FOC, 0.94 for WOC, and 0.86 for DWI. Positive predictive values were significantly higher on WOC images compared to all image types (p values <0.001to 0.006) except DWI (p = 0.151). There was no significant difference in PPV for WO compared to FO (p = 0.094) and FOC compared to FO (p = 0.683).

## Confidence score

The mean confidence scores were 2.9 for FO, 2.6 for WO, 2.7 for FOC, 2.9 for WOC and 3.2 for DWI (4-point Likert confidence scale: 1-no lesions, 2-indeterminate, 3-likely myeloma, 4-definitely myeloma).

Confidence scores were lowest for WO images, and FOC images, and not significant for WOC. They were, however, significantly higher for DWI images in comparison to other image types (p values from 0.000 to 0.004). Readers were more confident at identifying myeloma lesions on WOC compared to WO (p = 0.005). There was no statistically significant difference between FO and FOC images (p = 0.220).

## Sub-group analysis

Of 30 patients, 25 patients had focal lesions and 5 patients had diffuse disease. A summary of the mean lesion count, sensitivity, positive predictive value and confidence score for each image type for the diffuse disease group is given in Table 3 and the focal group in Table 4.

**Diffuse disease group.**

**Lesion count and sensitivity:** There was no significant difference in readers detecting diffuse disease between all image types.

There was higher sensitivity on DWI compared to FO images, but not significant (p = 0.07). There was no significant difference between WO, FOC, and WOC compared to FO (p = 0.855, p = 0.388 and p = 0.21 respectively).

**Positive Predictive Value and Confidence:** For positive predictive value there was no significance for WO, FOC, WOC or DWI compared to FO images (p = 0.934, p = 0.43, p = 0.135 and p = 0.423 respectively).

For confidence there was also no significance WO, FOC, WOC and DWI compared to FO images (p = 0.658, p = 0.768, p = 1, and p = 0.302 respectively).

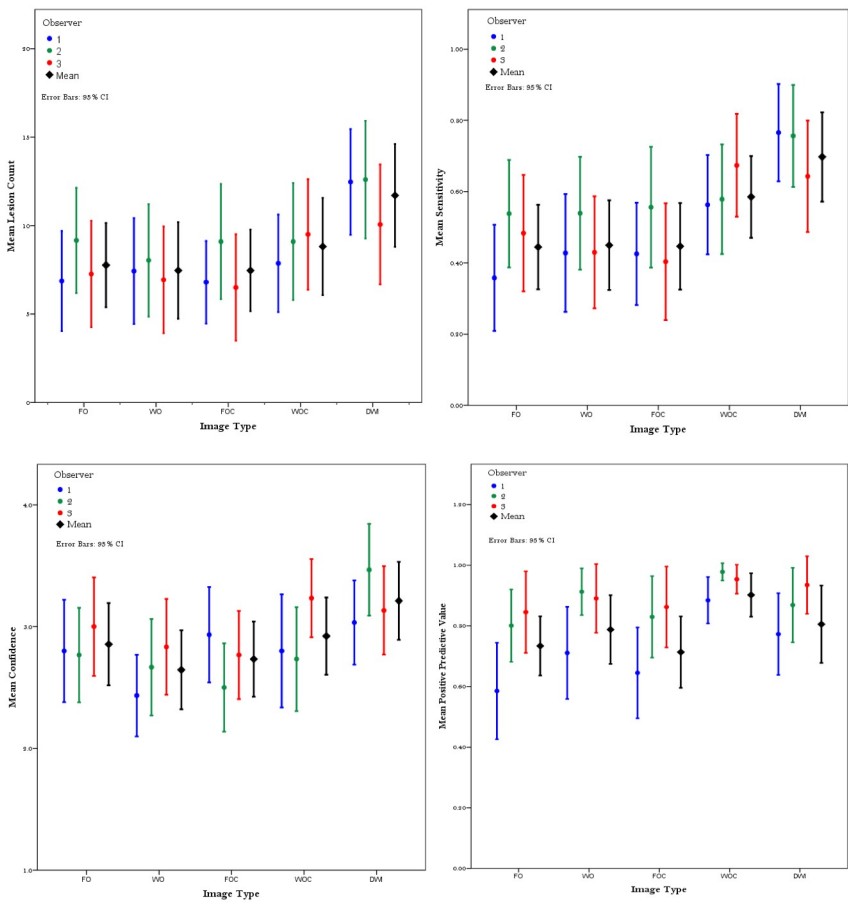

**Fig 3. Lesion count, sensitivity, positive predictive value (PPV) and confidence for each image type.** Individual observers are shown in colour (see legend), and the mean value across all three observers is shown in black. Error bars indicate the 95% confidence interval.

**Focal disease group.**

**Lesion count and sensitivity:** Lesions counts were significantly higher on DWI images compared to FO images in patients with focal disease (3.21, p = 0.001). There was no significant difference between the following image type comparisons FO vs WO, FOC and WOC (p = 0.826, p = 0.531, p = 0.349 respectively).

Sensitivity was significantly higher on DWI compared to FO images (4.02, p<0.0001). WOC images also demonstrated higher sensitivity compared to FO images (2.11, p = 0.04). WO and FOC images were not significantly different when compared to FO images (p = 0.715 and p = 0.754 respectively).

**Positive Predictive Value and Confidence:** Positive predictive value was significantly higher on DWI images compared to FO images (2.26, p = 0.024) and on WOC images compared to FO images (3.55, p<0.0001). No significant difference could be demonstrated on WO and FOC compared to FO (p = 0.055, p = 0.254, respectively).

For reader confidence DWI was significantly higher compared to FO images (2.16, p = 0.031), whilst there was so no significant difference of WO, FOC and WOC images compared to FO images (p = 0.256, p = 0.422 and p = 0.626 respectively).

**Patient by patient analysis of FO, WO, WOC, DWI:** As the above results show, DWI was significantly better than other image types for sensitivity, detecting true positive lesions and

**Table 3. Summary data for all patients.**

| All patients (n = 30) | | | | |
|---|---|---|---|---|
| **Lesion Count** | | | | |
| **Image type** | **Mean** | **Differences in means** | **(95% CI)** | **p-value** |
| FO | 7.8 | Baseline | - | - |
| WO | 7.5 | -0.37 | -1.9 to 1.3 | 0.713 |
| FOC | 7.5 | -0.37 | -1.9 to 1.3 | 0.713 |
| WOC | 8.8 | 1.29 | -0.55 to 2.66 | 0.196 |
| DWI | 11.7 | 4.83 | 2.34 to 5.55 | <0.001 |
| **Sensitivity** | | | | |
| **Image type** | **Mean** | **Differences in means** | **(95% CI)** | **p-value** |
| FO | 0.46 | Baseline | - | - |
| WO | 0.47 | 0.11 | -0.09 to 0.102 | 0.909 |
| FOC | 0.46 | 0.04 | -0.094 to 0.098 | 0.968 |
| WOC | 0.61 | 2.96 | 0.049 to 0.242 | 0.003 |
| DWI | 0.72 | 5.32 | 0.165 to 0.358 | <0.001 |
| **PPV** | | | | |
| **Image type** | **Mean** | **Differences in means** | **(95% CI)** | **p-value** |
| FO | 0.75 | Baseline | - | - |
| WO | 0.83 | 1.68 | -0.011 to 0.138 | 0.094 |
| FOC | 0.77 | 0.41 | -0.058 to 0.088 | 0.683 |
| WOC | 0.94 | 4.55 | 0.095 to 0.24 | <0.001 |
| DWI | 0.86 | 3.13 | 0.043 to 0.187 | 0.002 |
| **Confidence** | | | | |
| **Image type** | **Mean** | **Differences in means** | **(95% CI)** | **p-value** |
| FO | 2.9 | Baseline | - | - |
| WO | 2.6 | -2.12 | -0.41 to -0.02 | 0.034 |
| FOC | 2.7 | -1.23 | -0.32 to 0.07 | 0.22 |
| WOC | 2.9 | 0.67 | -0.13 to 0.26 | 0.503 |
| DWI | 3.2 | 3.57 | 0.16 to 0.55 | <0.001 |
| **True Positives** | | | | |
| **Image type** | **Mean** | **Differences in means** | **(95% CI)** | **p-value** |
| FO | 6.2 | Baseline | - | - |
| WO | 7.0 | 1.02 | -0.697 to 2.208 | 0.308 |
| FOC | 6.6 | 0.45 | -1.112 to 1.786 | 0.653 |
| WOC | 8.4 | 2.98 | 0.759 to 3.663 | 0.003 |
| DWI | 10.6 | 5.92 | 2.937 to 5.841 | <0.001 |

Lesion count, true positives, sensitivity, positive predictive value and confidence were compared between the five image types for all patients, using the fat only images as the baseline. Regression analyses used image types as the predictor variable, and lesion count/TP/sensitivity/confidence were used as the outcome variable. Mean values were calculated by the regression analysis and were equal to means calculated manually from all patients and all three radiologists.

reader confidence. Therefore, to ascertain whether other image types would add any further diagnostic information when detecting myeloma lesions in addition to DWI, a patient by patient analysis was undertaken for each observer. The image types examined were WOC, FO and WO as these had the highest values for sensitivity, true positive lesion rate and positive predictive value. An image type was assumed to add value if the reader observed more true positive lesions or fewer false positive lesions compared to DWI. In the 90 sets of comparisons (30 patients x 3 observers), WOC added value in 17, FO in 14 and WO in 6 out of 90 comparisons.

**Table 4. Summary data for the diffuse subgroup.**

| Diffuse disease (N = 6) | | | | |
|---|---|---|---|---|
| **Lesion Count** | | | | |
| Image type | Mean | Differences in means | (95% CI) | p-value |
| FO | 7.7 | Baseline | - | - |
| WO | 6.8 | -0.28 | -7.04 to 5.26 | 0.777 |
| FOC | 9.6 | 0.62 | -4.21 to 8.1 | 0.536 |
| WOC | 9.3 | 0.51 | -4.54 to 7.77 | 0.608 |
| DWI | 12.9 | 1.68 | -0.88 to 11.43 | 0.093 |
| **Sensitivity** | | | | |
| Image type | Mean | Differences in means | (95% CI) | p-value |
| FO | 0.337 | Baseline | - | - |
| WO | 0.314 | -0.14 | -0.34 to 0.30 | 0.855 |
| FOC | 0.43 | 0.55 | -0.23 to 0.41 | 0.582 |
| WOC | 0.48 | 0.92 | -0.17 to 0.47 | 0.357 |
| DWI | 0.65 | 2.93 | 0.11 to 0.56 | 0.07 |
| **PPV** | | | | |
| Image type | Mean | Differences in means | (95% CI) | p-value |
| FO | 0.74 | Baseline | - | - |
| WO | 0.73 | -0.08 | -0.25 to 0.27 | 0.934 |
| FOC | 0.64 | -0.79 | -0.36 to 0.15 | 0.43 |
| WOC | 0.93 | 1.49 | -0.06 to 0.45 | 0.135 |
| DWI | 0.86 | 0.8 | -0.15 to 0.36 | 0.423 |
| **Confidence** | | | | |
| Image type | Mean | Differences in means | (95% CI) | p-value |
| FO | 2.7 | Baseline | - | - |
| WO | 2.6 | -0.44 | -.91 to 0.57 | 0.658 |
| FOC | 2.67 | -0.29 | -0.85 to 0.63 | 0.768 |
| WOC | 2.78 | 0 | -0.74 to 0.74 | 1 |
| DWI | 3.17 | 1.03 | -0.35 to 1.12 | 0.302 |

Lesion count, sensitivity, positive predictive value and confidence were compared between the five image types for diffuse disease only, using the fat only images as the baseline. Regression analyses used image types as the predictor variable, and lesion count/sensitivity/confidence were used as the outcome variable. Mean values were calculated by the regression analysis and were equal to means calculated manually from all patients and all three radiologists.

## Reader timings

The average read time for each image type was: FO 98s, WO 77s, FOC 99s, WOC 84s and DWI 67s across all readers. When combined, total mean reading time for the 5 image types of the pelvis was 417s for a patient (averaged across all three readers). Compared to the baseline (FO), reading time was significantly faster for WO (7.15, p = 0.003), WOC (7.14, p = 0.012) and fastest for DWI (7.13, p<0.0001).

## Percent contrast and contrast-to-noise ratio

ROI analysis revealed that CNR was highest for DWI images at 65.7 (values for each image type were, FO 58.0, WO 34.7, FOC 32.7, WOC 29.9). It was significantly higher on DWI compared to WOC (p = 0.03), FOC (p = 0.049), WO (p = 0.05).

Percent contrast for ROIs was highest for WOC at 0.61 followed by DWI (0.51), FO (0.42), FOC (0.39) and lowest for WO (0.15). These data are charted in Fig 4.

**Table 5. Summary data for the focal subgroup.**

| Focal disease (N = 24) | | | | |
|---|---|---|---|---|
| **Lesion Count** | | | | |
| Image type | Mean | Differences in means | (95% CI) | p-value |
| FO | 7.8 | Baseline | - | - |
| WO | 7.6 | 0.22 | -2.13 to 2.67 | 0.826 |
| FOC | 6.9 | -0.63 | -3.17 to 1.63 | 0.531 |
| WOC | 8.7 | 0.94 | -1.25 to 3.55 | 0.349 |
| DWI | 11.4 | 3.21 | 1.53 to 6.33 | 0.001 |
| **Sensitivity** | | | | |
| Image type | Mean | Differences in means | (95% CI) | p-value |
| FO | 0.49 | Baseline | - | - |
| WO | 0.5 | 0.37 | -0.10 to 0.15 | 0.715 |
| FOC | 0.47 | -0.31 | -0.15 to 0.11 | 0.754 |
| WOC | 0.62 | 2.11 | 0.01 to 0.26 | 0.04 |
| DWI | 0.78 | 4.02 | 0.13 to 0.39 | <0.001 |
| **PPV** | | | | |
| Image type | Mean | Differences in means | (95% CI) | p-value |
| FO | 0.74 | Baseline | - | - |
| WO | 0.86 | 1.92 | -0.001 to 0.202 | 0.055 |
| FOC | 0.81 | 1.14 | -0.04 to 0.159 | 0.254 |
| WOC | 0.93 | 3.55 | 0.081 to 0.28 | <0.001 |
| DWI | 0.856 | 2.26 | 0.02 to 0.21 | 0.024 |
| **Confidence** | | | | |
| Image type | Mean | Differences in means | (95% CI) | p-value |
| FO | 2.9 | Baseline | - | - |
| WO | 2.7 | -1.14 | -0.05 to 0.14 | 0.256 |
| FOC | 2.8 | -0.8 | -0.47 to 0.2 | 0.422 |
| WOC | 3 | 0.49 | -0.25 to 0.41 | 0.626 |
| DWI | 3.2 | 2.16 | 0.03 to 0.69 | 0.031 |

Lesion count, sensitivity, positive predictive value and confidence were compared between the five image types for focal disease only, using the fat only images as the baseline. Regression analyses used image types as the predictor variable, and lesion count/sensitivity/confidence were used as the outcome variable. Mean values were calculated by the regression analysis and were equal to means calculated manually from all patients and all three radiologists.

## Discussion

In this study of whole body imaging for assessment of skeletal disease, lesion counts, true positive counts, sensitivity, positive predictive value, reader confidence and reading time were

**Table 6. Reader timings.**

| Reader Timings | | | | |
|---|---|---|---|---|
| Image type | Mean time (s) | Differences in Means | (95% CI) | P = value |
| FO | 99.38 | Baseline | - | - |
| WO | 77.77 | 7.15 | -35.76 to -7.61 | 0.003 |
| FOC | 97.17 | 7.07 | -16.51 to 11.2 | 0.705 |
| WOC | 82.65 | 7.14 | -31.88 to -3.89 | 0.012 |
| DWI | 60.36 | 7.13 | -53.28 to 25.32 | 0.000 |

Reader timings were compared between the five image types for all patients, using the fat only images as the baseline. Mean values were calculated by the regression analysis, and were equal to means calculated manually from all patients and all three radiologists.

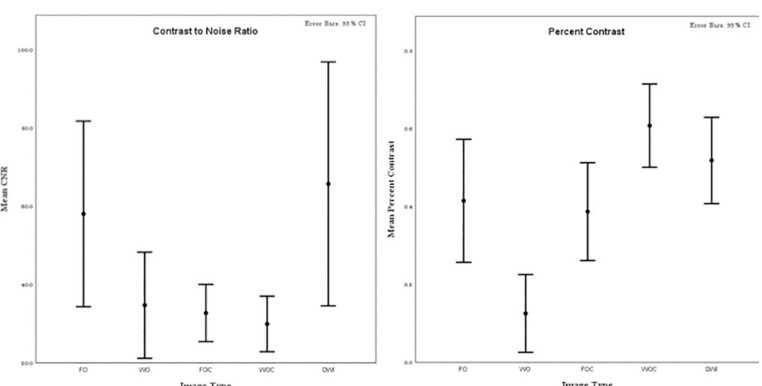

**Fig 4. Comparison of percent contrast and CNR between groups.** The figures show the results of a post- hoc multiple comparison test from a one-way ANOVA. Estimates of Percent Contrast and CNR are shown as circles; the comparison intervals for each group are shown as the whiskers.

compared across pre-contrast fat only (FO) and water only (WO) Dixon images, post contrast fat only (FOC) and water only (WOC) images, and diffusion weighted imaging (DWI). We have shown that DWI and WOC Dixon images are superior to other Dixon image types in detecting myeloma lesions. For all lesions combined (diffuse and focal disease) DWI proved to be the superior compared to all other image types for the following variables: lesion count, sensitivity, PPV, confidence and true positive. WOC was the next best image type with statistically significant differences in sensitivity, PPV and true positive counts compared to other image types apart from DWI. This was also reflected in the subgroup analysis which showed that for patients with focal disease, DWI and WOC were superior to the other image types for sensitivity and PPV. DWI performed best in detecting diffuse disease in all categories apart from PPV, however the results did not reach statistical significance. Lesion detection rates are highest as well as fastest (reading time in seconds) on DWI, followed by WOC images.

The observed bone lesion detection rates correlated with the results of the conspicuity analysis, which demonstrated that the DWI sequence had the highest contrast to noise ratio, and WOC had the highest percent contrast on ROI analysis. This is likely due to the highly cellular nature of myeloma lesions which causes replacement of normal fatty marrow [11,19], leading to an increase in signal on diffusion weighted imaging and enhancement of lesions against normal hypointense bone marrow on both DWI and WOC. There was no statistical difference between FO and FOC in lesion conspicuity and lesion detection rates. This suggests that enhancement of MM lesions on FOC does not improve conspicuity whereas on WOC, the hyperintense enhancement on the background of hypointense normal bone marrow leads to increased conspicuity and better detection. Both DWI and WOC image types provide functional information about myeloma lesions such as cellular density and vascularity leading to improved lesion contrast when compared to unenhanced Dixon images. This increase in conspicuity may be particularly important in myeloma patients with a higher bone marrow cell percentage in which lesions can be difficult to detect [11,20].

To the best of our knowledge, this is the first study to compare lesion detection rates on pre-contrast and post-contrast Dixon image types and DWI. The findings of this study are consistent with previous studies and recently published guidelines for the acquisition, interpretation and reporting of Whole-Body MRI in Myeloma (MY-RADS) [21]. Diffusion weighted imaging has been established as a sensitive sequence in detecting focal myeloma lesions compared to FDG-PET in a few studies [22,23] and forms a core component of the

recommended clinical protocol but suffers from the detection of false positive lesions. Dixon imaging is also recommended as a core component of imaging protocols for both anatomical and functional information such as fat fraction measurements [24]. The current guidelines recommend comparison with morphological imaging such as water only pre-contrast and fat fraction images to mitigate false positives detected on DWI. Our study has shown that post-contrast WOC is superior to WO and FO for sensitivity, positive predictive value and true positive detection rate. In addition, when a patient by patient analysis was undertaken, WOC added diagnostic value to DWI in 17 patients compared, WO (6) and FO (4). An image type was assumed to add value if the reader observed more true positive lesions or fewer false positive lesions compared to DWI, which would result in a change of management according to the International Myeloma Working Group (IMWG) guidelines. In patients with multiple myeloma lesions, detecting additional lesions may not change management but in patients with smouldering myeloma (SMM), detecting more than one unequivocal myeloma lesion puts patients in a high risk category leading to a change in management according to the IMWG guidelines [25]. Our results show that even the best performing image type (DWI) identifies fewer true positive lesions overall across all readers compared to the reference standard which included all image types. To optimise read times, a possible approach would be that radiologists assess post contrast WOC and DWI together using image linking and co-registration in order to detect truly positive lesions and reduce the false positives that DWI may detects on its own. This can be supplemented by review of the other imaging sequences, particularly where identifying a further lesion would impact on patient management. As commented above, identifying a second lesion in a patient with smouldering myeloma would have greater impact on disease management that identifying an additional lesion in a patient where multiple lesions were already seen. This reading strategy would reduce overall read times in some patients. The disadvantage of this approach is that it requires the administration of intravenous contrast which is associated with increased cost and the low risk of contrast reaction. However contrast administration also enables dynamic contrast enhanced imaging (WB-DCE) which has been shown to be of value in monitoring treatment [26,27] and as our study suggests would better reduce false positive lesions detected by DWI compared to unenhanced Dixon image types. We suggest that the choice of morphological imaging should be post contrast WOC in protocols which have post-contrast imaging and FO images in protocols which are unenhanced, in keeping with previous work and the value added shown by the patient by patient analysis [15].

In order to increase the value of MR, both acquisition time and read time need to be optimised. Our results show that for assessing the pelvis only, read times can be several minutes if all image types are assessed. If only the two most superior image types are assessed, the reading time saved would be 4 minutes 33 seconds (417s vs 143s) for a single image station. Whole body imaging in adults can require typically up to 7 stations, and the time saving for the whole scan read could potentially be in the order of tens of minutes if only 2 sequences are read rather than the entire datasets.

This study has some limitations. The reference standard included the image types that performed the best (DWI and WOC) and it may be argued that this can cause bias. However, the reference radiologist analysed all image types examined including in phase and out of phase Dixon images in addition to the five image types compared in this study, to determine a true positive lesion. Therefore, the potential bias is equal for all image types assessed and practically this represents the best possible reference standard available as the patients did not undergo another contemporaneous imaging modality (such as Pet-CT) and not all patients had equivalent follow up imaging. The scoring system used an upper limit of twenty lesions as the maximum lesions a reader detected. Therefore, in patients with high tumour load, we may not have

captured differences in lesion detection. The scoring system also penalised observers for failing to detect diffuse disease which is reflected in the low sensitivity scores compared to the reference standard. Another limitation of this study is that we have not compared lesion detection rates on other sequences which are used in other MM protocols such as STIR images. However, we feel that the time saving produced by Dixon imaging is sufficient justification for removing STIR imaging from WB-MRI protocols. Only six patients had a diffuse pattern of involvement and although DWI had higher indices than other image types, this did not reach statistical significance. This study has focussed on detection of bone disease and sequences required to assess extramedullary disease has not been considered. Reading times were recorded per image type. However in clinical practice, two or more types of images may be reviewed in parallel. Consequently simply summating reading times gives an indication of the potential time saving, but is not a precise measurement of time saved to read a whole scan. In our study we have not assessed the use of fat fraction maps as in clinical practice these are used for assessment of disease response during therapy, rather than for initial lesion identification.

In conclusion, the findings of this study suggest that DWI followed by WOC offer the highest relative value in terms of scan acquisition and read times, for bone lesion assessment in patients undergoing WBMRI for suspected myeloma. Our study suggests that WOC may be better than WO in supplementing DWI in the assessment of myeloma in treatment-naïve patients.

## Supporting information

**S1 File. Raw data showing lesion counts, true positives, false positives, false negatives, sensitivity, PPV and confidence for each of the four Dixon image types, for each observer and each subject.** Please refer to the Materials and Methods section for more information on data structure.
(XLSX)

## Acknowledgments

This work was undertaken at University College Hospitals NHS Trust and University College London. MHC, SP, TJPB and AL work within the National Institute for Health Research (NIHR) Biomedical Research Centre. The views expressed in this publication are those of the authors and not necessarily those of the NIHR, UK Department of Health. This work was supported by CRUK/EPSRC KCL/UCL Comprehensive Cancer Imaging Centre.

## Author Contributions

**Conceptualization:** Saurabh Singh, Timothy J. P. Bray, Arash Latifoltojar, Shonit Punwani, Margaret A. Hall-Craggs.

**Data curation:** Saurabh Singh, Elly Pilavachi, Alexandra Dudek, Timothy J. P. Bray, Arash Latifoltojar, Kannan Rajesparan, Margaret A. Hall-Craggs.

**Formal analysis:** Saurabh Singh, Elly Pilavachi, Alexandra Dudek, Timothy J. P. Bray, Kannan Rajesparan, Margaret A. Hall-Craggs.

**Funding acquisition:** Arash Latifoltojar, Shonit Punwani, Margaret A. Hall-Craggs.

**Investigation:** Saurabh Singh, Elly Pilavachi, Alexandra Dudek, Arash Latifoltojar.

**Methodology:** Saurabh Singh, Alexandra Dudek, Timothy J. P. Bray, Arash Latifoltojar, Kannan Rajesparan, Shonit Punwani, Margaret A. Hall-Craggs.

**Project administration:** Saurabh Singh, Elly Pilavachi.

**Resources:** Saurabh Singh, Elly Pilavachi, Alexandra Dudek.

**Software:** Saurabh Singh, Alexandra Dudek, Margaret A. Hall-Craggs.

**Supervision:** Saurabh Singh, Timothy J. P. Bray, Kannan Rajesparan, Shonit Punwani, Margaret A. Hall-Craggs.

**Validation:** Saurabh Singh.

**Visualization:** Saurabh Singh.

**Writing – original draft:** Saurabh Singh.

**Writing – review & editing:** Saurabh Singh, Elly Pilavachi, Alexandra Dudek, Timothy J. P. Bray, Margaret A. Hall-Craggs.

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
