## [Decision Letter · Decision Letter 0]

13 Nov 2019

PONE-D-19-26051

Whole Body MRI in Multiple Myeloma: Optimising Image Acquisition and Read Times

PLOS ONE

Dear Professor Hall-Craggs,

Thank you for submitting your manuscript to PLOS ONE. After careful consideration, we feel that it has merit but does not fully meet PLOS ONE’s publication criteria as it currently stands. Therefore, we invite you to submit a revised version of the manuscript that addresses the points raised during the review process.

We would appreciate receiving your revised manuscript by Dec 28 2019 11:59PM. To enhance the reproducibility of your results, we recommend that if applicable you deposit your laboratory protocols in protocols.io, where a protocol can be assigned its own identifier (DOI) such that it can be cited independently in the future. For instructions see: http://journals.plos.org/plosone/s/submission-guidelines#loc-laboratory-protocols

We look forward to receiving your revised manuscript.

Kind regards,

Pascal A. T. Baltzer, M.D.

Academic Editor

PLOS ONE

Journal Requirements:

2. We noticed you have some minor occurrence(s) of overlapping text with the following previous publication(s), which needs to be addressed:

https://doi.org/10.1371journal.pone.0180562

In your revision ensure you cite all your sources (including your own works), and quote or rephrase any duplicated text outside the Methods section. Further consideration is dependent on these concerns being addressed.

3. We have noticed that you report some instances of p = 0. Since this is not strictly possible, please report the exact p value in your results.

4. We noted in your submission details that a portion of your manuscript may have been presented or published elsewhere: The cohort of patients has been studied in previously published work. Two publications by Latifoltojar et al in 2017 studied 21 out of 30 patients in the context of changes in quantitative imaging biomarkers in response to treatment. A previous study by Bray et al 2017, studied the detection of myeloma lesions on pre-contrast imaging in this cohort. This study differs from both as we studied all available imaging including DWI and post contrast Dixon imaging in the entire 30 patient cohort. We have also included new patient by patient analysis, not performed in the paper by Bray et al 2017.  We have also focussed on MR value. We have measured read times for the scans for different readers, allowing estimation of time saving if scans are selectively reviewed..

Reviewers' comments:

Reviewer's Responses to Questions

**Comments to the Author**

1. Is the manuscript technically sound, and do the data support the conclusions?

Reviewer #1: Yes

Reviewer #2: Yes

2. Has the statistical analysis been performed appropriately and rigorously? 

Reviewer #1: Yes

Reviewer #2: Yes

3. Have the authors made all data underlying the findings in their manuscript fully available?

Reviewer #1: Yes

Reviewer #2: Yes

4. Is the manuscript presented in an intelligible fashion and written in standard English?

Reviewer #1: Yes

Reviewer #2: Yes

5. Review Comments to the Author

Reviewer #1: A complete full body protocol is used as a reference. In comparison, the individual sequences are analyzed sequentially. They are examined in detail with regard to their diagnostic accuracy. Finally, it is examined whether the combination of the best with the less important sequence offers an incremental benefit. This approach is generally convincing and worthy of being published. This is all the more true, since the data obviously come from a database that has already been published, which is, however, sufficiently noted.

A few points of criticism I would like to mention which should be addressed by the authors.

1. It should be made clear that this is an artificially shortened protocol. In fact, the full state-of-the-art protocol includes more sequences (https://doi.org/10.1148/radiol.2019181949, figure 1). In this respect, the reference standard is somewhat misleading and the actual accuracy of the sequences is presumably lower.

2. The authors mainly focus on parameters of lesion detection (sensitivity and PPV etc.). Just as important - especially in the context of efficiency - are the negative ratings. As far as I can see, these can also be analyzed by the data, which in my opinion should be revised.

3. The terminology efficiency and effectiveness is clearly defined, highly relevant, but unfortunately - in my opinion - not outlined clearly enough in the paper. In fact, the authors convincingly demonstrate that the shortening of the full protocol produces a significantly worse diagnostic result: E.g. lesion count / sensitivity of DWI = 72 %, which means a 28% worse detection rate vs. the gold standard. This, in turn, was not measured completely (see above). Of course, it is impossible to examine all this in one paper, but it should be named more clearly and discussed more objectively. This is all the more true in the context of the clinical relevance: The exact definition of the lesion load is vital for optimized outcome (etc.) as mentioned by the authors.

4. "Fat and water image reconstructions are mandatory and should be used to generate fat fraction maps": this clear statement in the above-mentioned cited paper is somewhat contradictory to the authors' conclusions. This, too, should be quite discussed.

5. The analysis of reading time is highly relevant and should be clearer presented. In every multiparametric analysis, one sequence is read as a "main sequence" and the other parameters are evaluated as add-ons. A simple addition of the reading times as given by the independent read is thus misleading in a certain way, but at least it overestimates the potential time saving effect.

6. CNR analysis: How were lesions not detectable in all sequences analyzed. A more detailed explanation is recommended.

7. Discussion: “increased (...) morbidity for patients” introduced by contrast agents. Unclear sentence. In which context side effects of CA could play a clinically relevant role in the management of MM patients?

8. Since MRI is a multiparametric method and will probably remain so, according to the data of the authors, a more detailed presentation of the data regarding the combination of several sequences is desirable ("determine which image types add diagnostic value").

9. Bland Altman analysis of the key results is recommended for better assessment of a proportional or systematic bias.

10. The main result is potential time savings for a different MRI protocols. I recommend to present these results more clearly in a graphical illustration.

Reviewer #2: Dear authors,

i have read your paper with great interest as there is only few data on evaluating specific sequences in WB/MRI in MM. I have found methodology, reporting and the conclusions drawn concise and comprehensible. I would like to ask you to comment on a. if there was any histopathologic correlation or consequent work-up by PET/CT in these 30 patients and

b. if the readers have had ADC maps when evaluation DWI.

Overall my recommendation would be: minor revisions.

With my best regards

6. PLOS authors have the option to publish the peer review history of their article (what does this mean?). If published, this will include your full peer review and any attached files.

Reviewer #1: No

Reviewer #2: No

---

## [Author Response · Author response to Decision Letter 0]

13 Dec 2019

Journal Requirements:

The guidelines have been reviewed and followed. 

2. We noticed you have some minor occurrence(s) of overlapping text with the following previous publication(s), which needs to be addressed:

https://doi.org/10.1371journal.pone.0180562

In your revision ensure you cite all your sources (including your own works), and quote or rephrase any duplicated text outside the Methods section. Further consideration is dependent on these concerns being addressed.

Outside the methods (and we have acknowledged and referenced the previous paper [15]), there is very little overlapping text as the results are of a different study.

All sources have been cited. 

3. We have noticed that you report some instances of p = 0. Since this is not strictly possible, please report the exact p value in your results.

The p values recorded as p = 0.000 have been amended to p<0.001. 

4. We noted in your submission details that a portion of your manuscript may have been presented or published elsewhere: The cohort of patients has been studied in previously published work. Two publications by Latifoltojar et al in 2017 studied 21 out of 30 patients in the context of changes in quantitative imaging biomarkers in response to treatment. A previous study by Bray et al 2017, studied the detection of myeloma lesions on pre-contrast imaging in this cohort. This study differs from both as we studied all available imaging including DWI and post contrast Dixon imaging in the entire 30 patient cohort. We have also included new patient by patient analysis, not performed in the paper by Bray et al 2017. We have also focussed on MR value. We have measured read times for the scans for different readers, allowing estimation of time saving if scans are selectively reviewed..

I think that the covering letter quoted from above was confusing and misleading. The paragraph below has been expanded to make it clear that the data submitted in the current study are new. Apologies for the confusion. 

“The CURRENT SUBMITTED study differs from both (the previous studies ) as we studied all available imaging including DWI and post contrast Dixon imaging in the entire 30 patient cohort. We have also included new patient by patient analysis, not performed in the paper by Bray et al 2017. We have also focussed on MR value. We have measured read times for the scans for different readers, allowing estimation of time saving if scans are selectively reviewed.”

We confirm that the new data included in the submitted paper have not been published elsewhere and this is not duplicate publication. 

Response to Comments to the Author

Reviewer #1: A complete full body protocol is used as a reference. In comparison, the individual sequences are analyzed sequentially. They are examined in detail with regard to their diagnostic accuracy. Finally, it is examined whether the combination of the best with the less important sequence offers an incremental benefit. This approach is generally convincing and worthy of being published. This is all the more true, since the data obviously come from a database that has already been published, which is, however, sufficiently noted.

A few points of criticism I would like to mention which should be addressed by the authors.

1. It should be made clear that this is an artificially shortened protocol. In fact, the full state-of-the-art protocol includes more sequences (https://doi.org/10.1148/radiol.2019181949, figure 1). In this respect, the reference standard is somewhat misleading and the actual accuracy of the sequences is presumably lower.

In the paper referred to by the reviewer, a different protocol was used to the one described in our study. We do not routinely image the spine, but if there is any abnormality seen in the spine on the Dixon and diffusion images, we go on to do dedicated spine images as an additional procedure. This saves time and is supported by data from the paper from our group authored by Latifoltojar (reference 16).

We do routinely do T2 images, but the main reason for doing the T2 is to assess for extramedullary disease and for treatment response following chemotherapy. In the current paper we have not included the T2 images as it well established in the literature, and in general practice, that for patients before treatment the T2 images are non-contributory for the assessment of bone lesions over and above diffusion and Dixon images. 

We have clarified in the text the use of T2 images. 

“Although other sequences were acquired during the standard whole body protocol, these were not included for assessment as these are acquired mainly for assessment of extramedullary disease (axial T2) and for measurement of response to therapy (T2, ADC and fat fraction maps before and after treatment).” 

2. The authors mainly focus on parameters of lesion detection (sensitivity and PPV etc.). Just as important - especially in the context of efficiency - are the negative ratings. As far as I can see, these can also be analyzed by the data, which in my opinion should be revised.

In our cohort all our patients had a high clinical suspicion of myeloma and our current study was to compare the performances of different sequences and not the performance of whole body MRI. To do this we have to take a lesion and compare it on different sequences – we cannot take ‘no lesion’ [= true negative] and compare this between sequences. Consequently in this study we have no ‘true negatives’ as we are labelling lesions and not patients. We have true positives, false positives and false negatives. True negatives do not exist within this study. 

3. The terminology efficiency and effectiveness is clearly defined, highly relevant, but unfortunately - in my opinion - not outlined clearly enough in the paper. In fact, the authors convincingly demonstrate that the shortening of the full protocol produces a significantly worse diagnostic result: E.g. lesion count / sensitivity of DWI = 72 %, which means a 28% worse detection rate vs. the gold standard. This, in turn, was not measured completely (see above). Of course, it is impossible to examine all this in one paper, but it should be named more clearly and discussed more objectively. This is all the more true in the context of the clinical relevance: The exact definition of the lesion load is vital for optimized outcome (etc.) as mentioned by the authors.

We agree this is an important point. We have added a sentence to elaborate on why we think preferentially reading DWI and post contrast would lead to increased reporting efficiency in the Discussion

“This can be supplemented by review of the other imaging sequences, particularly where identifying a further lesion would impact on patient management. As commented above, identifying a second lesion in a patient with smouldering myeloma would have greater impact on disease management that identifying an additional lesion in a patient where multiple lesions were already seen. This reading strategy would reduce overall read times in some patients.”

4. "Fat and water image reconstructions are mandatory and should be used to generate fat fraction maps": this clear statement in the above-mentioned cited paper is somewhat contradictory to the authors' conclusions. This, too, should be quite discussed.

We do understand this point and agree that it is a relative limitation of our study. However fat fraction maps most useful in a clinical setting for measuring disease response to therapy, and our study was about identifying lesions in disease and not quantifying response to treatment. We have added a paragraph to clarify this. 

“In our study we have not assessed the use of fat fraction maps as in clinical practise these are used for assessment of disease response during therapy, rather than for initial lesion identification.”

5. The analysis of reading time is highly relevant and should be clearer presented. In every multiparametric analysis, one sequence is read as a "main sequence" and the other parameters are evaluated as add-ons. A simple addition of the reading times as given by the independent read is thus misleading in a certain way, but at least it overestimates the potential time saving effect.

We agree this is a limitation of the way reading time was measured. However as each individual image type was being assessed, we did not use a ‘’locked sequential read”. We have added a sentence expanding on this within the limitations section. 

“Reading times were recorded per image type. However in clinical practice, two or more types of images may be reviewed in parallel. Consequently simply summating reading times gives an indication of the potential time saving, but is not a precise measurement of time saved to read a whole scan.” 

6. CNR analysis: How were lesions not detectable in all sequences analyzed. A more detailed explanation is recommended.

For measurement of CNR we selected lesions which were visible on all sequences. This has been clarified in the manuscript. 

Percent Contrast and Contrast-to-Noise Ratio

In patients with at least three focal lesions greater than 5 mm in diameter, visible on all image types, percent contrast and contrast-to-noise ratio (CNR) were calculated using a previously described method (11).

7. Discussion: “increased (...) morbidity for patients” introduced by contrast agents. Unclear sentence. In which context side effects of CA could play a clinically relevant role in the management of MM patients?

We have altered this sentence as we agree that following the cases of NSF, and the introduction of new guidelines for administration of IV contrast, there should be no increase in morbidity other than contrast reactions. 

8. Since MRI is a multiparametric method and will probably remain so, according to the data of the authors, a more detailed presentation of the data regarding the combination of several sequences is desirable ("determine which image types add diagnostic value").

We agree that this would be interesting, but is beyond the scope of this study which was directed towards lesion detection. A different study would be needed to address diagnostic value/outcome analysis. 

9. Bland Altman analysis of the key results is recommended for better assessment of a proportional or systematic bias.

We do not feel that Bland Altman would be the appropriate test of our data. Bland Altman is fundamentally an analysis of agreement. We are using a reference ‘gold’ standard and so we can simply compare this with other sequences. 

10. The main result is potential time savings for a different MRI protocols. I recommend to present these results more clearly in a graphical illustration.

We have not really compared different MRI protocols in this study, we used one protocol only, and so it would be difficult to compare this graphically – as this is not fundamentally what we have measured. We have looked at lesion detection and read time for different sequences, and these are presented numerically in the text. 

Reviewer #2: Dear authors,

i have read your paper with great interest as there is only few data on evaluating specific sequences in WB/MRI in MM. I have found methodology, reporting and the conclusions drawn concise and comprehensible. I would like to ask you to comment on a. if there was any histopathologic correlation or consequent work-up by PET/CT in these 30 patients and

b. if the readers have had ADC maps when evaluation DWI.

Overall my recommendation would be: minor revisions.

Thank you for your positive review. 

The patients had their bone marrow examined but focal lesions were not biopsied.

The readers did not have ADC maps to review although these were automatically produced by the vendor software. As with most read protocols, the long b value diffusion images are used for lesion detection (the focus of this study) whereas ADC maps and change in ADC are used for assessment of treatment response.

---

## [Decision Letter · Decision Letter 1]

15 Jan 2020

Whole Body MRI in Multiple Myeloma: Optimising Image Acquisition and Read Times

PONE-D-19-26051R1

Dear Dr. Hall-Craggs,

We are pleased to inform you that your manuscript has been judged scientifically suitable for publication and will be formally accepted for publication once it complies with all outstanding technical requirements.

With kind regards,

Pascal A. T. Baltzer, M.D.

Academic Editor

PLOS ONE

Additional Editor Comments (optional):

Reviewers' comments:

Reviewer's Responses to Questions

**Comments to the Author**

1. If the authors have adequately addressed your comments raised in a previous round of review and you feel that this manuscript is now acceptable for publication, you may indicate that here to bypass the “Comments to the Author” section, enter your conflict of interest statement in the “Confidential to Editor” section, and submit your "Accept" recommendation.

Reviewer #2: All comments have been addressed

2. Is the manuscript technically sound, and do the data support the conclusions?

Reviewer #2: Yes

3. Has the statistical analysis been performed appropriately and rigorously? 

Reviewer #2: Yes

4. Have the authors made all data underlying the findings in their manuscript fully available?

Reviewer #2: Yes

5. Is the manuscript presented in an intelligible fashion and written in standard English?

Reviewer #2: Yes

6. Review Comments to the Author

Reviewer #2: Dear authors,

i am happy with the manuscript as it is right now and i have no further questions.

Best regards.

7. PLOS authors have the option to publish the peer review history of their article (what does this mean?). If published, this will include your full peer review and any attached files.

Reviewer #2: No

---

## [Editor Report · Acceptance letter]

23 Jan 2020

PONE-D-19-26051R1 

Whole Body MRI in Multiple Myeloma: Optimising Image Acquisition and Read Times 

Dear Dr. Hall-Craggs:

I am pleased to inform you that your manuscript has been deemed suitable for publication in PLOS ONE. Congratulations! Your manuscript is now with our production department. 

With kind regards,

on behalf of

Dr. Pascal A. T. Baltzer 

Academic Editor

PLOS ONE